# Single-Cell Classification Based on Population Nucleus Size Combining Microwave Impedance Spectroscopy and Machine Learning

**DOI:** 10.3390/s23021001

**Published:** 2023-01-15

**Authors:** Caroline A. Ferguson, James C. M. Hwang, Yu Zhang, Xuanhong Cheng

**Affiliations:** 1Department of Bioengineering, Lehigh University, Bethlehem, PA 18015, USA; 2Department of Materials Science and Engineering, Cornell University, Ithaca, NY 14853, USA; 3Department of Materials Science and Engineering, Lehigh University, Bethlehem, PA 18015, USA

**Keywords:** broadband impedance, single-cell analysis, machine learning, support vector machine, microfluidics

## Abstract

Many recent efforts in the diagnostic field address the accessibility of cancer diagnosis. Typical histological staining methods identify cancer cells visually by a larger nucleus with more condensed chromatin. Machine learning (ML) has been incorporated into image analysis for improving this process. Recently, impedance spectrometers have been shown to generate all-inclusive lab-on-a-chip platforms to detect nucleus abnormities. In this paper, a wideband electrical sensor and data analysis paradigm that can identify nuclear changes shows the realization of a single-cell microfluidic device to detect nuclei of altered sizes. To model cells of altered nucleus, Jurkat cells were treated to enlarge or shrink their nucleus followed by broadband sensing to obtain the S-parameters of single cells. The ability to deduce important frequencies associated with nucleus size is demonstrated and used to improve classification models in both binary and multiclass scenarios, despite a heterogeneous and overlapping cell population. The important frequency features match those predicted in a double-shell circuit model published in prior work, demonstrating a coherent new analytical technique for electrical data analysis. The electrical sensing platform assisted by ML with impressive accuracy of cell classification looks forward to a label-free and flexible approach to cancer diagnosis.

## 1. Introduction

Diagnostic efforts to interrogate intracellular properties at a single-cell level traditionally require the extraction of intracellular materials like genetic material [1,2,3] or extracellular vesicles [4,5,6]. Other methods introduce stains for the purpose of identifying intracellular characteristics through fluorescence imaging or flow cytometry [7,8]. While all these methods boast merits, most are generally incompatible with the analysis of live, functional cells due to the necessary permeability to introduce stains or collect large quantities of intracellular material and take considerably longer than electrical analysis. While electrical measurement is a well-established paradigm for the identification of properties in both single cells and tissues, limitations exist in the available information electrical sensors can provide to the internal characteristics of living cells [9,10]. For rapid detection in a large cell population, impedance cytometry applications are typically limited to frequencies below the MHz range, limiting the ability to sense intracellular characteristics. This is because the cell impedance most heavily depends on the size and shape of the cells at the kHz range, and the cytoplasmic impedance manifests at the MHz or higher frequency ranges [11]. While several cytometric applications have been shown to identify cell population characteristics using MHz frequencies, these are typically dependent on the physical location of the cell with respect to the electrodes during measurement [12,13,14].

Broadband spectroscopy by contrast relies on cell trapping and positional consistency for the duration of the frequency sweep for accuracy up to GHz frequencies. With broadband electrical sensor designs and more complex data analysis methods including our previous work, subcellular features such as the nucleus or ionic property changes can be interrogated by GHz or microwave range signal measurements [15,16,17,18]. These approaches hold promises in single-cell cancer diagnosis with low resource requirements. Cancer nuclei are often larger with less chromosomal organization and a denser collection of chromatin within, properties that are typically determined by histological staining. However, the infrastructure to perform this procedure makes it difficult to implement in resource-limited areas [19]. While other cellular changes are associated with tumor cell expression, nucleus size is one of the most easily distinguishable by dielectric parameters and broadband spectroscopy, as shown in previous applications. By applying a label-free, lab-on-a-chip technology based on broadband electrical sensing to this clinical challenge, this work strives to offer an alternative way to look at nucleus size with fewer requirements on human operation and infrastructure.

Machine learning (ML) has been applied to clinical cancer identification based on nucleus properties in both genetic and histological image data to improve the speed and accuracy of both diagnosis and prognosis [20,21]. ML offers the benefit of incorporating various techniques to determine the most important features and appropriate diagnosis or prediction algorithms based on the problem at hand and being adaptable for larger datasets. For examination of single cells, recent studies have built classification models, such as support vector machine (SVM), gradient boosting (GB), k-nearest neighbor (KNN), and random forest (RF) with cell micrographs to identify the differentiation within a population of cells [22,23]. Similarly, apoptotic cell identification has been applied to single cells using fluorescence-based flow cytometry employing KNN, RF, SVM, and logistic regression (LR) classifiers [24]. The use of lab-on-a-chip set-ups combined with ML poses an interesting solution for examining overlapping populations of single cells which may be difficult to distinguish using traditional statistical analysis. This combination also promises to improve system integration, wherein clinical diagnosis could become more rapid, accurate, and require less resources. In recent years, this combination between ML and electrical data has been increasingly applied to classification tasks in various cell studies including cell type identification [25,26,27], cancer cell typing [28,29], and monitoring cell viability during treatment [30]. While SVM and LR remain some of the most commonly seen classifiers, typically a variety is used in each study considering the performance of classifiers is particularly dataset dependent [31]. Such classifier improvement can also be controlled using feature selection techniques and tuning parameters such as the regularization parameter in SVM or LR [32,33]. Especially in cases of high feature dimensionality, several recent diagnostic sensors using an e-nose paradigm emphasize the importance of feature selection methods to improve ML classification accuracy to refine the variables collected in future designs [34,35]. 

In the reported method, single cells are non-invasively measured using an ultra-broadband electrical sensor to successfully interrogate chemically induced changes to simulate both nucleus shrinkage associated with cell apoptosis and enlargement associated with cancer development. While previous work using broadband spectroscopy has already established the intracellular properties of apoptotic nuclei using a double-shell model [18], chemically enlarged nuclei simulating cancer characteristics have not been. Using this on-chip, broadband sensing approach, a variety of unique spectra obtained serve as the foundation for the ML approach. The comparison of feature selection methods and classifiers helped us identify patterns and frequencies of note in these varied impedance spectra. Using both traditional statistics and ML methods for feature reduction, this work examines the impact of a smaller feature set on classification accuracy using simple classification schemes. Here, we demonstrate the promising capability of leveraging microwave spectroscopy and ML to predict the specific nucleus treatment applied based on the resulting spectral feature patterns. In this paper, we present a new combination of feature selection, machine learning, and wideband spectral data to identify important frequencies between 9 kHz and 9 GHz to classify nucleus size changes. This work is an important step towards a paradigm for evaluation and analysis of clinical cells to determine label-free disease-related differences in cell impedance, which could prove especially useful for difficult-to-diagnose disease states.

## 2. Materials and Methods

Human lymphocyte cells from the Jurkat line (ATCC TIB-152) were cultured in RPMI 1640 supplemented with 10% FBS and 5% Penicillin/Streptomycin antibiotics. The cells were kept at 5% CO_2_ and 37 °C for the duration of their culture before treatment. Cells were separated from the culture flask and treated for 1 h using a final concentration of 1 mM Staurosporine (ST, Sigma Aldrich S6942, St. Louis, MO, USA) in cell culture media to shrink the nucleus. To enlarge the cell nucleus, cells were incubated overnight for 48 h using Ciprofloxacin (CP, MP Biomedicals 199020, Santa Ana, CA, USA) dissolved into PBS at a concentration of 10 µM. For measurement, cells were moved to an 8.5% sucrose and 0.3% dextrose solution in DI water to provide a low ionic sensing environment at a concentration of 1 × 10^6^ cells/mL. In previous work, we have shown the viability of cells in this solution for multiple hours of electrical testing. The nuclear size change was verified using fluorescence staining the nucleus using Hoechst 33342 (Invitrogen H3570) and staining the cytoplasm using CalcienAM (Biotium 80011-3) added to general culture media. From these fluorescence images, FIJI/ImageJ was used to extract both the cell and nucleus diameter, followed by an analysis of variance statistics to determine significant size changes.

Cells travel through a PDMS channel to pass directly over the surface of an electrode series gap where they are electrically trapped and then measured. The electrical sensing utilizes a 10 µm wide series gap between 100 µm width flat electrodes deposited in gold on a patterned coplanar waveguide (CPW). The PDMS channel is designed with two branches sandwiching the straight channel to provide a sheath force to direct cells to the center of both the channel and the electrodes. To prepare the entire cell capture device, the PDMS channel was attached to the CPW by aligning the channel center with the center of the series gap on the CPW. A sheath flow of 0.25 µL/min and a sample flow rate of 0.1 µL/min introduced the cell suspension to the CPW surface and guided them to the center of the electrode gap. To stop cells within the series gap, a dielectrophoretic signal of −9 dBm at a frequency of 500 MHz was applied using a Virtual Network Analyzer (Keysight ENA Network Analyzer E5080A). Following the isolation and stationary position of a single cell within the gap, a probing signal of −15 dBm sweeping over 201 frequencies from 9 kHz to 9 GHz was passed through the cell while recording the resulting impedance in two ports, resulting in 402 unique frequency features per cell. Each cell had the consistency of size and location validated through optical photographs using a microscope (Nikon).

At each frequency, two power ratios were measured, S11 and S21, or power reflection and power transmission to determine the proportion of power reflected or passing through. Directly after, the S11 and S21 of the sucrose solution alone were measured for comparison with the cell-influenced signal. A more universal parameter of ΔS11 or ΔS21 was calculated from subtraction between the two to see the small dB change caused by the cell presence between the electrodes.

The ML framework was built around a binary classification system examining different combinations of feature selection strategies and classifiers to identify informative feature patterns in distinguishing untreated (UNT), ST- and CP-treated populations of Jurkat cells with normally sized, shrunk, or enlarged nuclei. Initially, the data was normalized along the 402 frequency features using the electrical spectra for each treatment type. Two feature selection methods were separately implemented to reduce features based on either a statistical elimination (SE) scheme or recursive feature elimination (RFE). In the case of RFE, a minimum of 50 features was set as a parameter for the cross-validated based on the results of a sweep to optimize this parameter (Figure A1). For statistical elimination, a student’s *t*-test was used for binary classification pairs and features were removed when considered insignificant (*p* > 0.05) when comparing the classes. The reduced feature set was then split using 10-fold cross-validation for classification. Specifically, all cell samples were randomly divided into 10 folds, such that each fold was left out and used as a test set once while the remaining nine folds were used as a training set for the classifier training. In each run of the cross-validation, z-score normalization was applied to standardize each feature across samples. The classification was tested and compared using a linear kernel-based SVM and L2 regularized LR. This cross-validation was repeated 10 times, so each overall metric of accuracy, sensitivity, and specificity was averaged across 100 runs of a specific classification scheme for a reliable performance evaluation.

For multiclass classification, the resulting binary classifiers and selected features from the previous paragraph were each applied to a randomly selected subset of cells. For the subset of cells, normalization occurred by frequency feature and for each binary pair, the selected features were used to generate a classification prediction. A major voting strategy was employed, selecting the final multiclass classification based on which class received the most votes for that cell out of all three binary classifiers.

## 3. Results

### 3.1. Optical Characterization of Cell and Nucleus Size

To verify the treatments with Staurosporine (ST) and Ciprofloxacin (CP), the same batch of cells was split into groups with either no treatment (UNT) or one type of treatment. Using FIJI software, the cytoplasm and nucleus diameters were measured using fluorescent imaging, and the distribution of the cell diameter and the nucleus-to-cell ratio were each fitted to a normal distribution as visualized in Figure 1. The UNT Jurkat cells exhibited an average cell size of 10.5 µm overall diameter with a nucleus-to-cell-diameter ratio of 0.80 (Figure 2). Based on the measured diameters, analysis of variance (ANOVA) statistical testing was used to determine a significantly lower nucleus ratio of 0.77 (*p* = 0.025) without a significantly lower cell size (*p* = 0.227) for ST treatment. Similarly, for CP treatment, the nuclear size ratio was significantly larger at 0.83 (*p* = 0.0001) without significantly impacting the cell size (*p* = 0.494). These results verified that despite comparable cell sizes among the three populations, the average nucleus characteristics of the populations changed significantly with treatments. While our previous work has the capability of distinguishing ST treatment to elicit shrinking reactions in the nucleus [13], this verification of nuclear size increase using CP treatment created a more radical comparison of how nucleus size impacts electrical signals without causing cell death.

### 3.2. Visualization of Electrical Spectra

Upon collection of S-parameter spectra from UNT (n = 53), ST (n = 43), and CP-treated cells (n = 39) using the microfluidic flow device outlined in our earlier work [15,16,17], the spectra were compared among the two treatment conditions and the untreated cells. Figure 3 shows the overlap in the mean spectra (lines) and differences in standard deviation (shadows) at each frequency between the treated and untreated groups. While there is a clear population overlap between UNT, ST, and CP-treated cells, there are still certain characteristics that can be visually identified in the spectra that are specific to treatment. For example, in ST-treated cells, there is a shift of the peak to a higher frequency around 9 GHz of ΔS11 along with a decreased slope in the 10-MHz range of ΔS21, shown in red in Figure 3a,b, respectively. Alternatively, CP-treated cells show a deeper dip at 1 GHz of ΔS11 and an increased slope in the 10-MHz range of ΔS21 appearing blue in Figure 3. Based on previous circuit modeling focusing on how double-shell model compartment properties can impact different portions of spectra, these changes are consistent with increased resistance in the ST-treated cells and decreased resistance in CP-treated cells for both the nucleoplasm and cytoplasm [18]. For ST-treated cells, the changes are also consistent with the spectra changes seen in the previous work, and the current work measuring a much larger number of cells to allow ML-based classification.

Looking at the variety of spectra collected, it is obvious that statistical analysis methods to visually identify distinguishing features would be limited in their ability to accurately describe the changes associated with different treatments. The individual response to treatment in the cell populations is heterogeneous: both the nuclear size change seen during imaging and the impedance spectra of the three groups (UNT, ST, and CP) have overlapping features. Thus, it is not immediately apparent which frequency measurements were experiencing either the largest changes or changing most consistently. Although it is qualitatively apparent that most changes between the treated spectra occur at frequencies near to and greater than 1 GHz, there are also a variety of peak and valley features in the MHz frequencies that demand more sophisticated methods of comparison. Using machine learning classification, the properties of the entire spectra for each treatment label have been evaluated more comprehensively. Despite the heterogeneous population overlap masking the classes of treatment, these spectral changes attributing to the nucleus size change can be used to classify individual cells as belonging to a specific class of treatment, as described below.

### 3.3. Binary Feature Selection and Classification

For the purposes of determining the most important features, the ΔS11 and ΔS21 parameters were combined for a single feature set consisting of both measurements at each frequency. One of the unique challenges associated with feature selection in a dataset containing both ΔS11 and ΔS21 spectra is the vast scale differences between the S-parameter values. As seen in Figure 3, ΔS21 can range from 1.0 to 0.1 dB at higher frequencies, while the peak values of ΔS11 are on the order of 0.001 dB. To combat the tendency to select higher magnitude ΔS21 features, normalization across the two distinct parameter feature sets was applied prior to combination and feature selection, encouraging decisions based on the difference at each frequency value for either measurement type rather than the overall signal magnitude. The feature set of cells for each binary treatment class pair was reduced by either SE or RFE. For SE, traditional methods were used to compare the means of the populations at each frequency using a student’s t-test to determine a significant difference in signal mean at each frequency among the classes. Frequencies with a *p*-value < 0.05 were kept in the dataset, while the others were removed. RFE made selections by weighting feature importance and eliminating based on the resulting accuracy from a simplified classifier model. Either method resulted in a combination of frequencies at which ΔS11 and ΔS21 values are most relevant for improved classification accuracy.

The resulting reduced frequency set underwent typical normalization across individual cells before being classified by both SVM and LR using scikit-learn [36], as demonstrated in Figure 4a. The frequency weight pattern for each binary pair is consistent with the visual identifiers discussed previously, especially using recursive feature elimination (RFE) (Figure 4b,c). From previous knowledge, cytoplasmic properties are easier to identify and dominate the signal at MHz frequencies, and they often dominate the spectra properties by order of magnitude [18]. These frequencies also show significant changes in treated populations in this work as the ratio of cytoplasm to the whole cell is changing with the nucleus shrinking (ST) or expanding (CP). Based on a heatmap showing the classifier weights associated with each frequency after RFE feature selection, the ΔS11 signal has the most important features in the GHz range, where the peaks are observed (Figure 4b). Looking more closely, there is a red band used by the SVM classifier around 6 GHz for both pairs including UNT cells, right around where the peak is typically seen. Similarly, in the GHz range for pairs including CP weighted by the SVM classifier, there are blue bands where the dip was seen in many ΔS11 spectra. For the ΔS21 spectra, the MHz frequency measurements were the most heavily weighted in all cases where the slope of the signal is clearly different based on the sample spectra (Figure 4c). Another notable distinction in Figure 4c is the high GHz ΔS21 signal weights selected for pairs including ST, suggesting that although small in magnitude, the change of the ΔS21 signal in the GHz range is consistently associated with the nucleus size reduction in the heterogeneous population.

With the pairwise comparisons of the treatment types, we observed that the overall highest accuracy without any feature selection was seen for the ST/CP classification due to the most extreme difference in population nucleus size. The effect of both statistical difference and feature selection can be seen in the difference exhibited in error bars in Figure 4d. Here we see that the error bars become smaller as the difference between the nucleus size groups becomes more distinct, for example, the difference between UNT/CP was greater than UNT/ST, with the biggest difference being between the two treated groups. Similarly, the error bar size becomes smaller in the case of feature selection as it improves both the magnitude and consistency of the prediction.

The overall accuracy success in the ST/CP set is consistent with the largest nuclear size difference expected between those two populations. Although feature selection improves the classification accuracy, the most consistently improved classification for each pairwise comparison was RFE combined with SVM (Figure 4d). The RFE + SVM combo is especially effective at predicting either ST or CP cell signals from the UNT cell signals, thereby distinguishing if some type of treatment has taken place in the cell population. The treatment distinction is especially relevant considering the heterogeneous nature of the treated cell populations and the tendency of overlapping signal characteristics of single cells. However, at this point it should be noted for the cancer diagnostic purposes inspiring this work, the most relevant result is the UNT/CP classification with excellent accuracy of up to 94% derived using the RFE + SVM combination. Looking at the distribution of predictions themselves for this binary combination, the analysis further clarifies that SVM boasts a more balanced and robust prediction set for the UNT/CP classification scenario (Figure A2). Using the RFE findings, the feature weights for these trained spectra were examined by pairs to determine which frequencies were most crucial in ΔS11 and ΔS21 to distinguish nucleus size.

### 3.4. Multiclass Prediction

To build upon the success of the binary classification schemes, a major voting approach was developed based on the predictions each binary classifier would produce for each cell in the dataset. While the same overall structure was maintained, the predictions of each classifier were considered to generate a final multiclass prediction if each cell was UNT-, ST-, or CP-treated, as shown in Figure 5a. The prediction process using major voting was completed for each combination of feature selection method and classifier, showing more success in the case of RFE than SE (Figure 5b). In this case, the error bars are contingent on the predictions made by the binary classifiers but the same pattern of reduced error bar size with feature selection is observed. The multiclass prediction method exemplified that using feature selection to focus on relevant features as the basis of classification, most of which occur in the 100-MHz to 9-GHz range, there can be a 96% classification accuracy among all three classes, especially in identifying ST- or CP-treated cells (Figure 5c). The major voting approach also showed much higher prediction accuracy when compared to training a multiclass classifier among the three classes.

The outlined approach is unique in analyzing the individual spectra over broadband frequencies, containing information on the membrane, cytoplasm, and nucleus conditions. Literature reported SVM classifiers using electrical data based on two or three frequency measurements, typically in the kHz and low MHz ranges, limiting the amount of feature information distinguishing the classes [37,38,39,40]. By collecting 402 features in a broad frequency range and determining informative ones by the classifiers in a data-driven manner, the sensing paradigm has the flexibility to function not only in cancer diagnosis but in identifying properties associated with different tissue and disease types. Traditional techniques of diagnosis rely on specific biomarkers or optical properties, but a more generalized electrical and feature-tuned classifier approach can be adapted and validated on multivariate disease-related changes. Going forward, the validation of the system and classifier will require measurements on both a new and independent dataset of cells with nuclei varying in size due to other treatments and eventually, application to clinical samples of tumors with nucleus abnormalities.

## 4. Discussion

The advantage of broadband sensors in nucleus-based diagnostic applications lies in the ability to examine larger populations of cells rapidly while parsing individual properties. This would become particularly useful when both cancerous and noncancerous cells exist in a collected population. By measuring single cells, the distinction between two heterogeneous and overlapping cell populations containing distinctly different cells can be used to fine-tune our understanding of internal compartment electrical properties however slight the signal change. While similar sensors for rapid quantification and property interrogation rely on simulation-fitted parameters [28,41], this work shows that using appropriate feature selection, these results can be matched through raw spectra analysis in terms of prediction accuracy.

Based on a review of existing literature for stationary cell broadband spectroscopy, most classification from similar sensors for single cells is done by training the classifier on modeled parameters of internal cell characteristics like cytoplasm capacitance and permittivity rather than the raw spectra data to improve overall model accuracy [26,28,29]. Classification then relies on the data collection, initial fitting to cell parameters using a circuit model, and the classifier accuracy itself, introducing multiple levels of models needed, potential error, and computational burden. To combat the need for this multi-step modeling, we compared different combinations of feature reduction and popularly used classifiers with raw spectra measurements to identify the best methods using both ΔS11 and ΔS21. In this subsection, the classifier combination frequency feature weights are shown to be comparable to previous circuit fitting results without requiring them during the analysis itself. This more rapid and circuit-free approach will be especially translatable when identifying a cell population from clinical samples containing both cancerous and normal cells, but without previous knowledge of the cellular changes.

Many other works that apply machine learning to electrical impedance data on single cells rely on impedance cytometry measurements or impedance spectroscopy on a few frequencies [42,43]. While these methods typically have the advantages of larger sample sizes as measurements are faster and in the case of cytometry, the cells move continuously. However, the method shared shows the application on a larger number of frequencies and higher broadband frequencies than are shown in other works. By examining the features as well as classification, the information in this could be applied to further design or tune devices that rely on a limited number of frequencies for measurement or classification. The feature selection method helps re-establish the most important frequency features for determining the difference between the populations using a data-driven method rather than purely circuit modeling, which could help tune future limited-frequency high-throughput impedance spectrometry experiments

Similar work completed by Joshi et al. showed the ability of a quadratic discriminant analysis (QDA)-based classification model combined with a microfluidic and electrical micro-impedance cytometer to differentiate between the healthy breast tissue and two types of blood cancer cells [28]. Their study trained a classifier using features of the impedance, current, and phase information fit from a single-shell electrical model, requiring additional computational steps, and not interrogating any specific properties to differentiate between the cells. The paradigm in this work can directly correlate our classifiers selected features to nucleus size change based on (1) population size measurements and (2) a matching double-shell electrical model with separate compartments for both the cytoplasm and nucleus size. As mentioned previously, sweeping different feature selection and classification methods make this set-up adaptable to sense not just nucleus size but also other associated properties without requiring the additional computational step of fitting intracellular properties before classification. In contrast, the system of prediction discussed in this work measures a wide range of frequencies and information, letting feature selection on training data determine which are optimal to include for both binary and multiclass classification systems. While circuit modelling helps determine and explain the internal properties changing, the feature selection can help guide this process and focus such computationally heavy methods. Although the presented method is early in development, it shows capability to identify nucleus enlargement in a single-cell microfluidic device in a way that could be extended to more extensive clinical phenotyping in the future. There is also ongoing work to extend this detection system to other diseases.

## 5. Conclusions

In this study, we examined the combination of microwave impedance spectroscopy and ML classification to identify the electrical properties of Jurkat cells with different nuclei-altering treatments. It is shown that these nucleus treatments significantly change overall nucleus size in a heterogeneous population which is reflected in minute changes to the high-frequency power spectra of single cells. Using a pairwise binary class combination of spectra consisting of 402 frequency features, a unique approach is demonstrated showing feature selection utilization to improve classifier accuracy. The resulting analysis demonstrates the ability to (1) identify relevant frequencies even in overlapping cell populations without circuit modeling and (2) train binary classifiers with high accuracy to improve multiclass prediction. The most successful binary classification scheme shows the ability to identify nucleus size increases in untreated cells, which is clinically relevant in diagnosing cancer from healthy tissue. Similarly, an excellent multiclass accuracy was obtained through a major voting strategy enabling the prediction of treatment for each sample. Based on the findings in this paper, work moving forward will include trying similar methods on different nucleus disease models and clinical samples from data collected on the same electrical device. Expansion to clinical samples of different tumor phenotypes is expected to show the applicability of this paradigm in identifying cancerous cells based on multivariate electrical changes where physical resources or frequency of symptom presentation are scarce.

## Figures and Tables

**Figure 1 sensors-23-01001-f001:**
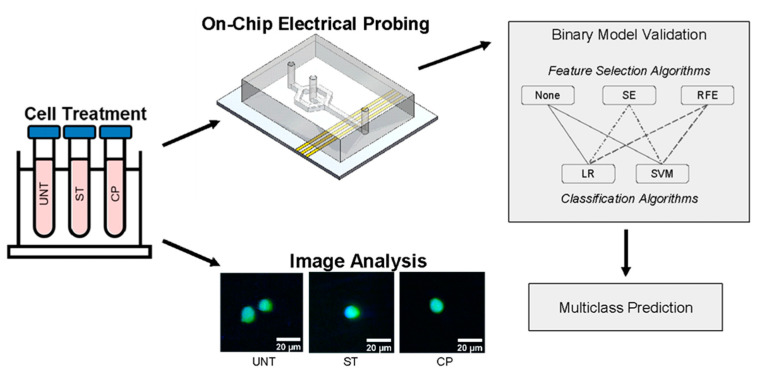
Experimental structure overview demonstrating both data collection and application of the ML paradigm of dual normalization, feature selection, and classification.

**Figure 2 sensors-23-01001-f002:**
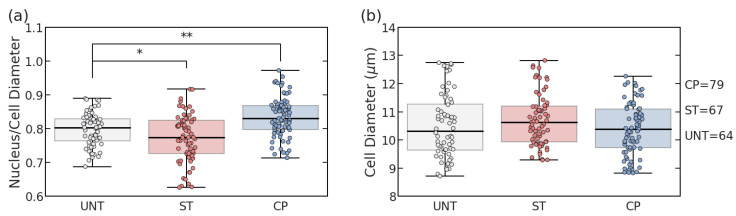
Cell and nucleus diameter measurements from fluorescence imaging. (**a**) Nucleus/cell diameter ratios among untreated, staurosporine (ST), and ciprofloxacin (CP) treated Jurkat cells. (**b**) Population cell diameter comparison among treatment types. (* *p* < 0.05, ** *p* < 0.01) Each box demonstrates the interquartile range (IQR) for the sample and the whiskers show the extension to 1.5 * IQR.

**Figure 3 sensors-23-01001-f003:**
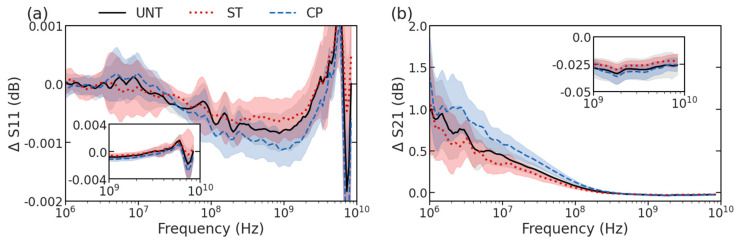
Swept spectra showing mean (lines) and standard deviation (shadows) for (**a**) ΔS11 and (**b**) ΔS21. UNT cells are shown using black with a gray shadow, ST−treated cells are shown using red line and shadow, and CP−treated cells are shown using a blue line and shadow.

**Figure 4 sensors-23-01001-f004:**
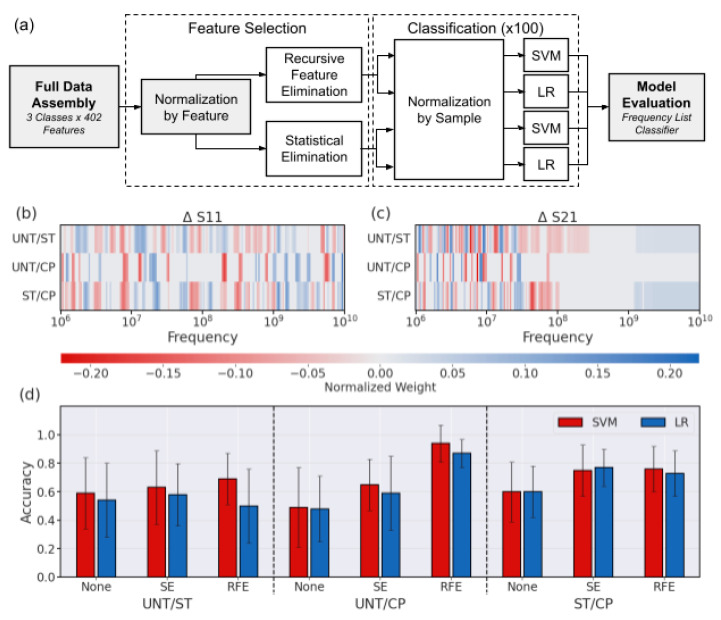
Approach and evaluation of cross-validated pairwise binary classifiers. (**a**) Schematic describing binary classification paradigm structure. (**b**) ΔS11 and (**c**) ΔS21 feature weights based on RFE + SVM classification for each pair. (**d**) Accuracy for predictions using each feature selection method and classifier for all binary pairs. The error bar indicates the standard deviation across the runs of cross-validation.

**Figure 5 sensors-23-01001-f005:**
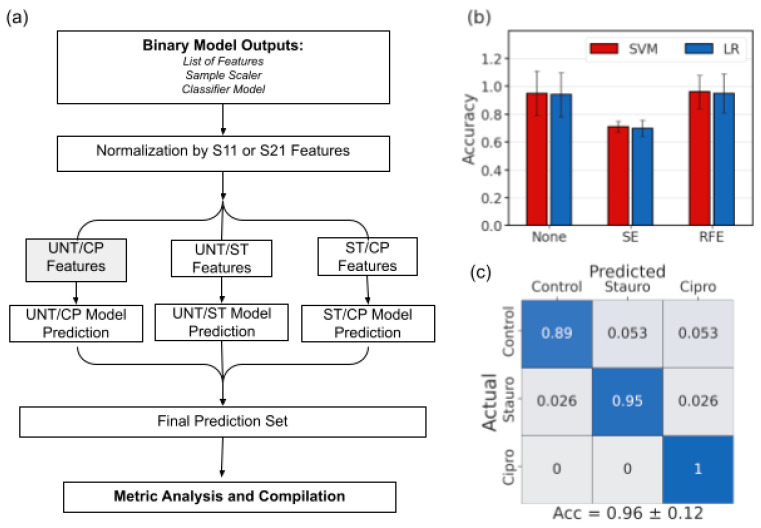
Multiclass classification approach using major voting and previous binary classifier results. (**a**) Schematic describing the multiclass voting structure. (**b**) Final classification accuracy of the multiclass predictions for 10 randomly selected cells. (**c**) Confusion matrix describing prediction percentages for each cell type.

## Data Availability

Not applicable.

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
