# Peer review of "Single-Cell Classification Based on Population Nucleus Size Combining Microwave Impedance Spectroscopy and Machine Learning"

_sensors, 2023, doi:10.3390/s23021001_

Round 1
Reviewer 1 Report
1) Please write the novelty of this work in Introduction section.
2) Did the authors apply this method in real tumor cells?
3) Discuss the accuracy of the method in both binary and multiclass scenarios. (why the error bar is high in Figure 4?)
4) compare your work with other published works. (advantages and disadvantages)
Reviewer 2 Report
In this manuscript, the authors describe a wideband electrical sensor and data analysis paradigm combining of microwave impedance spectroscopy and machine learning classification to identify the electrical properties of Jurkat cells with different nuclei altering treatments. It is shown that these nucleus treatments significantly change overall nucleus size in a heterogeneous population which is reflected in minute changes to the high-frequency power spectra of single cells. Using a pairwise binary class combination of spectra consisting of 402 frequency features, a unique approach is demonstrated showing feature selection utilization to improve classifier accuracy. The electrical sensing platform assisted by ML with impressive accuracy of cell classification which is expected to provide new ideas for unmarked and flexible cancer diagnosis methods. The manuscript is well-organized and clearly stated. I would suggest accepting it after the following minor concerns are addressed:
1、In the introduction, the author introduces the combination between ML and electrical data applying classification tasks. Please specify the advantages or effects of such combination on speed or accuracy.
2、The explanation in Figure 1 about “SE”, “RFE ” ,“LR” and “ SVM” has been mentioned in the previous article, so it is unnecessary to repeat the explanation.
3、 In in Figure 2, please specify how many cells are tested in the experiment to get the results shown.
4、Line 185-188, the author mentioned the work in Reference 13, which proved that ST and CP can change the size of the nucleus, and they will not cause cell death with electrical testing. However, this view is not directly mentioned in that reference. Is there a more powerful experiment or reference to prove this view?
Round 2
Reviewer 1 Report
The authors have addressed all the comments, and, and, and the manuscript has been improved after the revision. Now, it can be publicated in "Sensors".